# Synthesis, Characterization, and Application of Carboxymethyl Cellulose from Asparagus Stalk End

**DOI:** 10.3390/polym13010081

**Published:** 2020-12-28

**Authors:** Warinporn Klunklin, Kittisak Jantanasakulwong, Yuthana Phimolsiripol, Noppol Leksawasdi, Phisit Seesuriyachan, Thanongsak Chaiyaso, Chayatip Insomphun, Suphat Phongthai, Pensak Jantrawut, Sarana Rose Sommano, Winita Punyodom, Alissara Reungsang, Thi Minh Phuong Ngo, Pornchai Rachtanapun

**Affiliations:** 1School of Agro-Industry, Faculty of Agro-Industry, Chiang Mai University, Chiang Mai 50100, Thailand; warinporn.k@cmu.ac.th (W.K.); jantanasakulwong.k@gmail.com (K.J.); yuthana.p@cmu.ac.th (Y.P.); noppol@hotmail.com (N.L.); phisit.s@cmu.ac.th (P.S.); thachaiyaso@hotmail.com (T.C.); chayatip@yahoo.com (C.I.); suphat.phongthai@cmu.ac.th (S.P.); 2The Cluster of Agro Bio-Circular-Green Industry (Agro BCG), Chiang Mai University, Chiang Mai 50100, Thailand; 3Center of Excellence in Materials Science and Technology, Chiang Mai University, Chiang Mai 50200, Thailand; pensak.amuamu@gmail.com (P.J.); sarana.s@cmu.ac.th (S.R.S.); winitacmu@gmail.com (W.P.); 4Department of Pharmaceutical Sciences, Faculty of Pharmacy, Chiang Mai University, Chiang Mai 50200, Thailand; 5Plant Bioactive Compound Laboratory (BAC), Department of Plant and Soil Sciences, Faculty of Agriculture, Chiang Mai University, Chiang Mai 50200, Thailand; 6Department of Chemistry, Faculty of Science, Chiang Mai University, Chiang Mai 50200, Thailand; 7Department of Biotechnology, Faculty of Technology, Khon Kaen University, Khon Kaen 40002, Thailand; alissara@kku.ac.th; 8Research Group for Development of Microbial Hydrogen Production Process, Khon Kaen University, Khon Kaen 40002, Thailand; 9Academy of Science, Royal Society of Thailand, Bangkok 10300, Thailand; 10Department of Chemical Technology and Environment, The University of Danang—University of Technology and Education, Danang 550000, Vietnam; ntmphuong@ute.udn.vn

**Keywords:** agricultural waste, asparagus, biopolymer, carboxymethyl cellulose, CMC, degree of substitution, DS, cellulose extraction

## Abstract

Cellulose from *Asparagus officinalis* stalk end was extracted and synthesized to carboxymethyl cellulose (CMC_as_) using monochloroacetic acid (MCA) via carboxymethylation reaction with various sodium hydroxide (NaOH) concentrations starting from 20% to 60%. The cellulose and CMC_as_ were characterized by the physical properties, Fourier Transform Infrared spectroscopy (FTIR), Differential scanning calorimetry (DSC), Scanning electron microscopy (SEM) and X-ray diffraction (XRD). In addition, mechanical properties of CMC_as_ films were also investigated. The optimum condition for producing CMC_as_ was found to be 30% of NaOH concentration for the carboxymethylation reaction, which provided the highest percent yield of CMC_as_ at 44.04% with the highest degree of substitution (DS) at 0.98. The melting point of CMC_as_ decreased with increasing NaOH concentrations. Crystallinity of CMC_as_ was significantly deformed (*p* < 0.05) after synthesis at a high concentration. The *L** value of the CMC_as_ was significantly lower at a high NaOH concentration compared to the cellulose. The highest tensile strength (44.59 MPa) was found in CMC_as_ film synthesized with 40% of NaOH concentration and the highest percent elongation at break (24.99%) was obtained in CMC_as_ film treated with 30% of NaOH concentration. The applications of asparagus stalk end are as biomaterials in drug delivery system, tissue engineering, coating, and food packaging.

## 1. Introduction

Asparagus (*Asparagus officinalis* L.) is a nutritious and perennial vegetable continually used as antifungal activities, anticancer and anti-inflammatory herbal medicine in Asia. In the processing of asparagus, the spears which are 2–3% of total weight of the asparagus are typically processed into three types of products: canned, fresh, and frozen [1]. The residues from the processing were used for animal feeding and produced low-value products due to the high content of cellulose [2]. Therefore, tons of asparagus stalk end are the agriculture by-product which can cause environmental pollution [2]. With abundant cellulose and photochemical properties of cellulose in asparagus by-products, it acts as a good source of new value-added products [1], including biological compounds and functions [3,4], a dietary fiber [5] and nanocellulose [6]. Agricultural by-products from asparagus are rich source of celluloses which can be isolated from asparagus stalk end. A little work has been carried out to extract the cellulose from the asparagus; however, the previous works have not comprehensively considered to synthesis and apply CMC from asparagus stalk end as a film or coating to a food industry.

Cellulose is the most abundant polymer present in the primary cell wall of plants, algae and the oomycetes [6,7]. The abundant cellulose consists of repeated links of β–D-glucopyranose which can convert into high value cellulose esters and ethers [7]. The cellulose is insoluble in water which limits many applications. To increase the utility of cellulose, the carboxymethylation process of cellulose is able to syntheses water-soluble cellulose derivatives called carboxymethylcellulose (CMC) [8]. The forecast for global carboxymethyl cellulose market is estimated to reach approximately USD 1.86 billion by 2025. Food market generated the highest revenue of the global CMC market in 2016 with a market share of 33.7% [9]. The productions of CMC synthesized from agricultural waste have been studied such as sago waste [10], *Mimosa pigra* peel [11], durian rind [8], cotton waste [12], corn husk [13], corncob [14], grapefruit peel [15], rice stubble [16], papaya peel [17], *Juncus* plant stems [18], and mulberry wastepaper [19].

Thus, CMC can be versatile applied in various field such as drug delivery carrier [20], wound healing application [21], active corrosion protection [22], binder in ceramics [23], composited in hydrogel film [24] and hydrogel food packaging [25], fresh fruit and vegetable coatings [26,27] and hydrocolloid in noodles, bakery products and other foods [28,29] and it can produce composite films. CMC films exhibit many worthwhile attributes than many other biopolymers such as biodegradability, ease of production, forming a stable cross-linked matrix for packaging, etc. However, no research has presented the production of CMC powder from asparagus stalk end (CMC_as_) and its application such as packaging film so far.

Therefore, this study aimed to determine the effect of NaOH concentration at 20–60% on percent yield, the degree of substitution (DS), thermal properties, chemical structure, viscosity, crystallinity, morphology of CMC_as_. The effect of NaOH concentration mechanical properties (tensile strength and percentage elongation at break) and morphology of CMC_as_ films was also carried out.

## 2. Materials and Methods

### 2.1. Materials

Asparagus stalk end was purchased from Nong Ngulueam Sub-District (Nakhon Prathom, Thailand). All chemicals used in the preparation and analysis of synthesized CMC were analytical reagent (AR) grade or the equivalent. Absolute methanol, ethanol, and sodium silicate from Union science Co., Ltd. (Chiang Mai, Thailand), hydrogen peroxide from QReC^TM^ (Auckland, New Zealand), sodium hydroxide and glacial acetic acid from Lab-scan, isopropyl alcohol (IPA), Monochloroacetic acid from Sigma-Aldrich (Steinheim, Germany).

### 2.2. Materials Preparation

Two grams of asparagus stalk end was cut into small pieces and dried in an oven 105 °C for 6 h. Then dried asparagus was weighed to calculate the percent dryness as described in Equation (1). Dried asparagus stalk end was grounded by using a hammer mill (Armfield, Ringwood, Hampshire, UK) to less than 1 mm. The dried powder was then stored in polypropylene (PE) bags at ambient temperature until used. The percent dryness was calculated by the following Equation (1)
(1)$$\%Dryness=\frac{Weight\;of\;cellulose\;without\;moisture\;content}{Weight\;of\;cellulose\;content}\times 100$$

### 2.3. Extraction of Cellulose from Asparagus Stalk End

Cellulose from dried asparagus stalk end powder was extracted according to the method of [11,16]. Briefly, the dried powder of asparagus stalk end was extracted with a ratio of cellulose to 10% (*w*/*v*) NaOH solution at 1:20 (*w*/*v*) treated at 100 °C for 3 h. The black slurry was filtered and rinsed with cold water until a neutral pH of rinsed water was obtained. To obtain the cellulose fiber, the washed fiber residue was dried in an oven at 55 °C for 24 h. Hydrogen peroxide and sodium silicate were used to bleach the cellulose fiber. The bleached cellulose was then grounded by using the hammer mill size 70 mash (Armfield, Ringwood, Hampshire, UK). The bleached cellulose powder was kept in PE bags at ambient temperature until used.

### 2.4. Carboxymethyl Cellulose (CMC) Synthesized from Asparagus Stalk End

50 mL of various concentrations of NaOH at 20, 30, 40, 50, and 60% (*w*/*v*) and 350 mL of isopropanol (IPA) were blended with 15 g of cellulose powder extracted from asparagus stalk end and this was mixed for 30 min. Subsequently, the carboxymethylation reaction was synthesized according to the method of [11]. The final CMC_as_ product was in powder form. The percent yield was calculated by the following Equation (2) [11]:
(2)$$Yield\;of\;CMC\;(\%)=\frac{Weight\;of\;CMC\;(g)}{Weight\;of\;cellulose\;(g)}\times 100$$


### 2.5. Determination of the Degree of Substitution (DS) of CMC_as_

The degree of substitution (*DS*) of CMC_as_ presents the average number of hydroxyl group replaced by carboxymethyl and sodium carboxymethyl groups at C2, 3, and 6 in the cellulose structure. The *DS* of CMC_as_ was evaluated by the USP XXIII method for Crosscarmellose sodium which are titration and residue on combustion [11,23]. Calculation of the *DS* was showed in the following Equation (3);
(3)DS=A+S 
where *A* is the *DS* of carboxymethyl acid and *S* is the *DS* of sodium carboxymethyl which *A* and *S* were calculated using Equations (4) and (5):(4)$$A=\frac{1150M\;content}{\left(7120-412M-80C\right)Content}$$
(5)$$S=\frac{(162+58A)C\;content}{\left(7120-80C\right)content}$$
where *M* is consumption of the titration to end point (mEq) and *C* is the amount of ash remained after ignition (%).

### 2.6. The Determination of Percentage of Residue on Ignition

A crucible was dried in an oven at 100 °C for 1 h and transferred to a desiccator until the weight reached at an accurate value (Scale, AR3130, Ohaus Corp. Pine, NJ, USA) [23]. CMC_as_ was added into a crucible. The crucible containing CMC_as_ were ignited at 400 °C using a kiln (Carbolite, CWF1100, Scientific Promotion, Sheffield, UK) for approximately l to l.5 h and placed into the desiccator to obtain black residue. Sufficient sulfuric acid was added to moisten the entire residues and heated up until the gas fumes (white smoke) was utterly disappeared. The crucible with the black residue was ignited at 800 ± 25 °C to produce white residue and placed in the desiccators to get a constant weight. The residue on ignition was presented in the percentage and calculated using Equation (6).
(6)$$The\;percentage\;of\;residue=\;\frac{Weight\;of\;residue\;content}{Weight\;of\;CMC\;content}\times 100$$

### 2.7. Fourier Transform Infrared Spectroscopy (FTIR)

The aldehyde groups of the cellulose from stalk asparagus and CMC_as_ were determined by using FTIR spectra (Bruker, Tensor 27, Billerica, MA, USA). 2 mg of dry sample was pressed into a pellet with KBr. Transmission level measured at the wavenumber range of 4000–400 cm^−1^ [28].

### 2.8. Viscosity of Cellulose and CMC_as_

A Rapid Visco Analyzer (RVA-4, Newport Scientific, Warriewood, Australia) was used to determine the viscosity of samples. 1 g of cellulose or CMC_as_ sample was weight and dissolved in 25 mL of distilled water with stirring at 80 °C for 10 min—automatic stirring action set at 960 rpm for 10 s. The temperature of the sample was varied from 30, 40, 50 and 60 °C at 5 min intervals and held speed at 160 rpm until end of the test [11].

### 2.9. Thermal Conductivity Measurements

The thermal characterization of cellulose from asparagus stalk end and CMC_as_ were determined using Differential Scanning Calorimeter (DSC) (Perkin Elmer, Kitakyushu, Japan) according to the melting temperature determined from the thermogram. The determination was performed under a nitrogen atmosphere at a flow rate of 50 mL min^−1^. Samples (10 mg) contained in aluminum pans were heated from 40 to 450 °C with a heating rate of 10 °C min^−1^ [11].

### 2.10. X-ray Diffraction (XRD)

The crystallinity of the samples was determined using X-ray diffraction patterns carried out by X-ray powder diffractometer (JEOL, JDX-80-30, Shimadzu, Kyoto, Japan) in the reflection mode. Prior to the test, the samples were dried in a hot air oven (Memmert, Büchenbach (Bavaria, Germany) at 105 °C for 3 h to produce a powder form. Scans were carried out in the range of scattering angle (2θ) from 10 to 60° with at a scan rate of 5°/min [13].

### 2.11. Scanning Electron Microscopy (SEM)

Scanning Electron Microscopy (SEM) (JEOL JSM-5910LV SEM; Tokyo, Japan) was used to analyze the surface morphology of cellulose from asparagus stalk end and CMC_as_. The samples were analyzed through a large field detector. The acceleration voltage was used at 15 kV with 1500× original magnification [12].

### 2.12. Color Characteristics

The color characteristic of cellulose form asparagus stalk end and CMC_as_ was evaluated by using a Color Quest XE Spectrocolorimeter (Hunter Lab, Shen Zhen Wave Optoelectronics Technology Co., Ltd., Shenzhen, China) in order to express the CIELAB color as three values: *L** for the lightness from blackness (0) to whiteness (100), *a** from greenness (−) to redness (+), and *b** from blueness (−) to yellowness (+). The total color differences (∆*E*) take into account the comparisons between the *L**, *a** and *b** value of the sample and standard and calculated by the following Equation (7) [8]
(7)$$∆E=\sqrt{{\left(L_{Standard}^{*}-L_{Sample}^{*}\right)}^{2}+{\left(a_{Standard}^{*}-a_{Sample}^{*}\right)}^{2}+{\left(b_{Standard}^{*}-b_{Sample}^{*}\right)}^{2}}$$

The whiteness index (*WI*) was also calculated by the following Equation (8) to represent the degree of whiteness of samples [30].
(8)$$WI=\sqrt{\left(100-L^{{*}^{2}}\right)+a^{{*}^{2}}+b^{{*}^{2}}}$$

### 2.13. Preparation of CMC_as_ Film

The film-forming solutions were prepared by dissolving 4 g of CMC_as_ in 180 mL of distilled water with a constantly stirred at 80 °C under magnetic stirring for 15 min. 15% (*w*/*w*) glycerol was added to the solution and stirred for additional 5 min. The solution was poured on acrylic plates (20 cm × 15 cm) and the dried at room temperature for 72 h. After that, the film was peeled from the plates and stored in polyethylene bags at ambient temperature [11].

### 2.14. Mechanical Properties of CMC_as_ Film

The thickness of CMC_as_ films was determined using a micrometer model GT-313-A (Gotech Testing Machine Inc., Taichung, Taiwan, China). The examined mechanical properties which were tensile strength (*TS*) and percent elongation at break (*EB*) of the CMC_as_ film were tested (10 measurement each) using a universal texturometer (Model 1000 HIK-S, UK) according to the standard procedure of ASTM D882-80a [31] with preconditioning for 24 h and determined at 27 ± 2 °C with a relative humidity (RH) of 65 ± 2% according to Thai industrial standard for oriented polypropylene film (TIS 949-2533). The rectangular CMC_as_ films were cut into 15 × 140 mm as test specimens. The specimens were carried out by using an initial grip separation distance of 100 mm and crosshead speed at 20 mm/min. The *TS* and *EB* were calculated by following Equations (9) and (10), respectively.
(9)$$TS\;\left(MPa\right)=\frac{The\;maximum\;load\;\left(N\right)}{Weight\;of\;each\;film\times Thickness\;of\;each\;film}$$
(10)$$EB\;\left(\%\right)=\;\frac{The\;length\;of\;the\;film\;rupture-The\;initial\;length\;of\;the\;film}{The\;initial\;length\;of\;the\;film}$$

### 2.15. Statistical Analysis

Statistic data were analyzed by a one-way analysis of variance (ANOVA) using SPSS software version 16.0 0 (SPSS, an IBM company, Chicago, IL, USA). Duncan’s multiple range test was employed to evaluate significant differences among the treatments (*p* < 0.05). All measurements were analyzed in triplicated. The results were represented the mean values ± standard deviation. The figures present the standard deviation as the appropriate values. The error bars for some data points overlap the mean values.

## 3. Results and Discussion

### 3.1. Percent Yield of Carboxymethyl Cellulose from Asparagus Stalk End (CMC_as_)

The percent yield of CMC_as_ synthesized with various NaOH concentrations at 20, 30, 40, 50 and 60% (*w*/*v*) is shown in Figure 1. The percent yield of CMC_as_ increased when increasing of NaOH concentration from 20% to 30% (*w*/*v*) and then decreased with further increasing NaOH concentration at over 30% (*w*/*v*). The resulted phenomenon might be occurred due to a limitation of sodium monochloroacetate (NaMCA) as etherifying agents for substituting cellulose. Moreover, the NaOH concentration was not high enough to complete for converting the cellulose into alkali cellulose [32]. This result was similar to the work of synthesis carboxymethyl cellulose from durian rind [8].

### 3.2. Degree of Substitution (DS) of CMC_as_

The effect of various concentrations of NaOH on DS of CMC_as_ was carried out shown in Figure 2. The DS of CMC_as_ was in between 0.49 and 0.98 which reached the highest value at the concentration of NaOH at 30% (*w*/*v*). However, the DS of CMC_as_ was reduced at higher concentration of NaOH at 40–60% (*w*/*v*) ranged from 0.83–0.49. This phenomenon can be explained by investigating the carboxymethylation process, where two reaction occur concurrently. The first reaction involved a cellulose hydroxyl reacting with sodium monochloroacetate (NaMCA) in the presence of NaOH to obtain CMC_n_ shown in Equation (11) and Equation (12).
R–OH + NaOH → R–ONa + H_2_O(11)
R–ONa + Cl–CH_2_–COONa → R–O–CH_2_–COONa + NaCl(12)

The second reaction involves NaOH reacting with NaMCA to form sodium glycolate as by-product shown in Equation (13) [33].
NaOH + Cl–CH_2_COONa → HO–CH_2_COONa + NaCl(13)

The second reaction overwhelms the first reaction together with a strong alkaline concentration. If the alkaline level in the second reaction is too high, a side reaction will form a high level of sodium glycolate as a by-product, thus lowering the DS. This result was agreed with the result found by [11] who studied CMC from *Mimosa pigra* peel. The degradation was taken place due to a high concentration of NaOH [11]. Moreover, similar results have been reported in [8]. The maximum DS value (0.98) of CMC from durian rind was also got from the NaOH concentration at 30% and this was also related to the trend in percent yield presented in Figure 1.

### 3.3. Fourier Transform Infrared Spectroscopy (FTIR) of Cellulose from Asparagus Stalk End and CMC_as_

FTIR was used to evaluate functional groups changes in the cellulose from asparagus stalk end and CMC_as_ structures. The substitution reaction of CMC_as_ during the carboxymethylation was confirmed by FTIR [28]. Cellulose and each CMC_as_ have similar functional groups with same absorption bands such as hydroxyl group (–OH stretching) at 3200–3600 cm^−1^, C–H stretching vibration at 3000 cm^−1^, carbonyl group (C=O stretching) at 1600 cm^−1^, hydrocarbon groups (–CH_2_ scissoring) at 1450 cm^−1^, and ether groups (–O– stretching) at 1000–1200 cm^−1^ [34]. The FTIR spectra of the cellulose and CMC_as_ synthesized with various NaOH concentrations are shown in Figure 3. The carbonyl group (C=O), methyl group (–CH_2_) and ether group (–O–) notably increased in the CMC_as_ sample; however, the absorption band of hydroxyl group (–OH) reduced when compared to those cellulose samples (Figure 3). This result confirmed that the carboxymethylation had been substituted on cellulose molecules [32,34].

### 3.4. Effect of Various NaOH Concentrations on Viscosity of CMC_as_

The relationship between different NaOH concentrations and viscosity of CMC_as_ solution is shown in Figure 4. The viscosity of the CMC_as_ solution (1 g/100 mL) reduced with increase in the temperature. Increasing temperature plays a role in reducing the cohesive forces while simultaneously increasing the rate of molecular interchange, causing the lower viscosity [11]. According to the literature, the viscosity of CMC_as_ solution affected by NaOH concentrations was changed in accordance with DS value at the same temperature [33,35]. The CMC_as_ synthesized with 20% (*w*/*v*) NaOH concentration was the highest in viscosity due to the swelling and gelatinization of cellulose comparable to previous studies [11]. As far as NaOH concentration rose above 20%, the viscosity fell down. The decreasing of viscosity at a higher NaOH concentration was because of the degradation of CMC polymer led to a lower DS value that provides a small number of hydrophilic groups reducing the ability of the polymer to bond among water molecules [32,36]. An increase of the DS value also increased the CMC’s ability to immobilize water in a system [36]. Nevertheless, the viscosity of CMC depends on many influencing factors such as solution concentration [32], pH value, NaOH concentrations [36], and temperatures [11].

### 3.5. Effect of Various NaOH Concentrations on Thermal Properties of CMC_as_

The effect of various NaOH concentrations on the thermal property of cellulose from asparagus stalk end and CMC_as_ are shown in Figure 5. The melting temperature (T_m_) of cellulose and CMC_as_ synthesized with 20, 30, 40, 50, and 60 % (*w*/*v*) NaOH concentrations were 178.79, 206.61, 193.13, 182.34, 178.34 and 161.21 °C, respectively. The highest T_m_ of CMC_as_ increased when 20% (*w*/*v*) NaOH concentration was employed. The T_m_ at 20–30% (*w*/*v*) of NaOH concentrations were higher than T_m_ of cellulose due to the substituent of carboxymethyl groups influenced on increasing ionic character and intermolecular bonds between the polymer chains [37]. The T_m_ at 40–50% (*w*/*v*) of NaOH concentrations decreased due to the melting temperature of CMC_as_ slightly decreased as increasing the concentration of NaOH because of the increase in alkalization reaction together with a high substitution of carboxymethyl group. Moreover, the decrease in T_m_ was caused by an increase in number of the carbon atoms producing carbon skeleton, after a rapid quenching cooling with rate of 80 °C/min from the isotropic state at 200 °C [38]. As the T_m_ of 60% (*w*/*v*) of NaOH concentration decreased, side reaction was in the majority with sodium glycolate forming as a by-product and chain breaking of CMC_as_ polymer similar to the results presented in *Mimosa pigra* peel [11].

### 3.6. X-ray Diffraction (XRD) of Cellulose from Asparagus Stalk End and CMC_as_

The changes on the structure of cellulose from asparagus stalk end and CMC_as_ were evaluated by using XRD shown in Figure 6. The strength of hydrogen bonding and crystallinity contribute to the microstructure of CMC_as_ material. All CMC_as_ samples were less value in a peak of intensity (au) compared to cellulose from asparagus stalk end. Decrease in crystallinity index of cellulose may be due to the reorganization or cleavage of molecular according to alkalization of NaOH solution [32]. The increased aperture among cellulose polymer molecules was also caused by the substitution of monochloroacetic acid (MCA) molecules into the hydroxyl group of cellulose macromolecules easier than the cellulose without treating with an alkali solution [23].

These results are accord with results of various studies which were tested in cavendish banana cellulose [32] and durian rind [8]. The crystallinity index was also decreased when alkalizing by 15% (*w*/*v*) NaOH [32]. Thus, the crystallinity of CMC_as_ decreased by the effect of alkali solution prior to the carboxymethylation reaction on cellulose structure.

### 3.7. Morphology of Cellulose from Asparagus Stalk End and CMC_as_

Morphology of cellulose from asparagus stalk end and CMC_as_ powder with different NaOH concentrations were characterized by SEM with an acceleration voltage of 15 kV and 1500× original magnification (Figure 7a–f). The surface of cellulose from asparagus stalk end (Figure 7a) showed a smooth without pores or cracks and with a dense structure; however, the appearance of small fibers has been found in the cellulose. The increase of NaOH concentration affected changes in microstructure of the samples. The morphology of CMC_as_ powder treated with 20% of NaOH concentration emerged small fiber with minimal damage (Figure 7b). By increasing the content of NaOH from 30% to 60% (*w*/*v*), the surface uniformity decreased. The surface of CMC_as_ powder mixed with NaOH concentration at 30% started to peel without cracking as shown in Figure 7c. The surface of CMC_as_ powder has been begun to crack and deform when the powder treated with 40% of NaOH concentration (Figure 7d), due to the degradation of CMC polymer chain. Increasing the NaOH concentration at 50% (Figure 7e), the morphology of CMC_as_ powder increased irregularity surface with more collapsed and indented. The surface was more roughly and totally deformed when CMC_as_ treated with a higher NaOH concentration. Thus, this phenomenon could be inferred that CMC_as_ was synthesized with a high NaOH concentration causing a damage in surface area of cellulose powder. This result was comparable with a synthesized carboxymethyl from rice starch [23] and cassava starch [39]. Alkaline solution probably reduced the strength of structure, loss the crystallinity and changed the stability of the molecular organization of the cellulose causing the etherifying agents to have more access to the molecules for the carboxymethylation processes [36]. Consequently, this result was according to DS value.

### 3.8. Color of Cellulose from Asparagus Stalk End and CMC_as_

Color measurement was carried out to evaluate the color formation caused by carboxymethylation reaction. The color values of CMC_as_ were decreasing in *L** value as NaOH concentrations increased from 20 to 60% (*w*/*v*) of NaOH concentration. The a* and b* values showed the highest increased in CMC_as_ treated with 40% (*w*/*v*) of NaOH concentration. Moreover, an increase in NaOH concentration up to 40% decreased the yellowness of CMC_as_ probably due to the first reaction of carboxymethylation (Equation (12)) to produce CMC or sodium glycolate [36]. At higher NaOH concentration (50 and 60%, *w*/*v*), the decreasing of all color values was occurred. Therefore, the carboxymethylation reaction might be the reason the color values of cellulose and CMC_as_ has been changed [35]. The ∆E of the cellulose and CMC_as_ had similar trend compared to a* and b* values. These results are akin to the results from the previous studies [23]. The WI of the cellulose and CMC_as_ had same trend with *L** value (Table 1).

### 3.9. Mechanical Properties of CMC_as_ Films

The tensile strength (TS) and percentage of elongation at break (EB) of CMC_as_ films synthesized with various NaOH concentrations were determined by using the ASTM Standard Method D882-80a (Table 2). The TS value increased gradually with increasing NaOH concentration up to 40% (*w*/*v*) of NaOH. The TS value was correlated with an increasing DS value due to the substituted carboxymethyl groups in anhydroglucopyranose unit affecting an increase in intermolecular force among the polymer chains and the ionic character [32]. Nevertheless, the TS value of CMC_as_ films started to decline at a high NaOH concentration due to the polymer degradation and formation of by-product generated from sodium glycolate increased the reduction of CMC content and thus decreasing the intermolecular forces.

The EB of CMC_as_ films also increased with increasing NaOH concentrations up to 30% (*w*/*v*) of NaOH and drop down after that. At high NaOH concentrations, the crystallinity of the cellulose structure decreased with increasing a flexibility. In addition, CMC_as_ films has been obtained the lower flexibility at 60% (*w*/*v*) of NaOH concentrations because of an occurrence of hydrolysis reaction in cellulose chain [36].

## 4. Conclusions

This study revealed that *Asparagus officinalis* stalk end could be favorably used as a raw material to produce carboxymethyl cellulose. The alkali solution is the main parameter that affected all characteristics of CMC_as_ synthesized with various NaOH concentrations. The percent yield of CMC_as_ initially increased with NaOH concentrations increase up to 30% (*w*/*v*) and then gradually decreased with further NaOH concentrations increase related with the DS value of CMC_as_. The lightness (*L**) of CMC_as_ has been changed after treated with increasing NaOH concentration. The crystallinity of CMC_as_ was found to decrease after synthesized with alkali solution. The highest tensile strength and elongation at break values of CMC_as_ films were found in 40% (*w*/*v*) NaOH-synthesized CMC_as_ films and 30% (*w*/*v*) NaOH-synthesized CMC_as_ films, respectively.

## Figures and Tables

**Figure 1 polymers-13-00081-f001:**
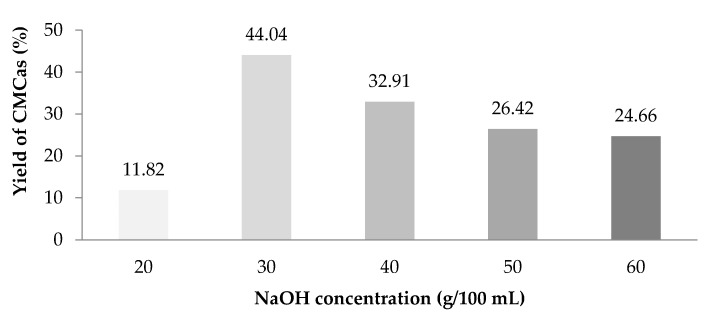
Percent yield of carboxymethyl cellulose from asparagus stalk end (CMC_as_).

**Figure 2 polymers-13-00081-f002:**
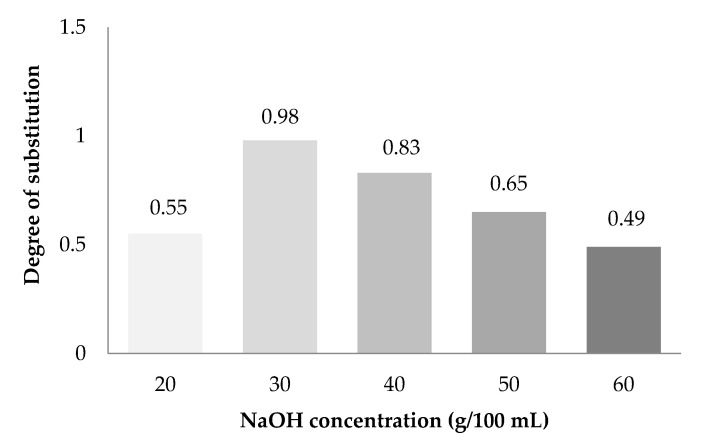
Effect of NaOH on degree of substitution of CMC_as_.

**Figure 3 polymers-13-00081-f003:**
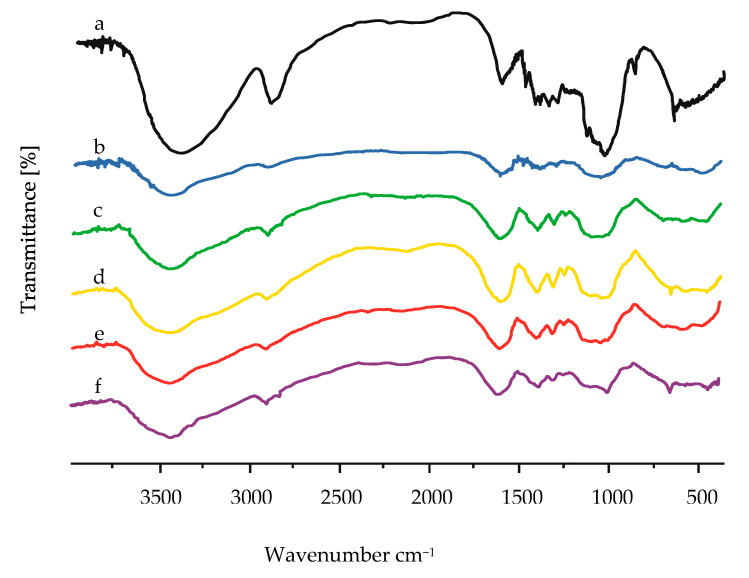
FTIR spectra of (a) cellulose, (b) CMC_as_ synthesized with 20% (*w*/*v*) NaOH concentration, (c) CMC_as_ synthesized with 30% (*w*/*v*) NaOH concentration, (d) CMC_as_ synthesized with 40% (*w*/*v*) NaOH concentration, (e) CMC_as_ synthesized with 50% (*w*/*v*) NaOH concentration and (f) CMC_as_ synthesized with 60% (*w*/*v*) NaOH concentration.

**Figure 4 polymers-13-00081-f004:**
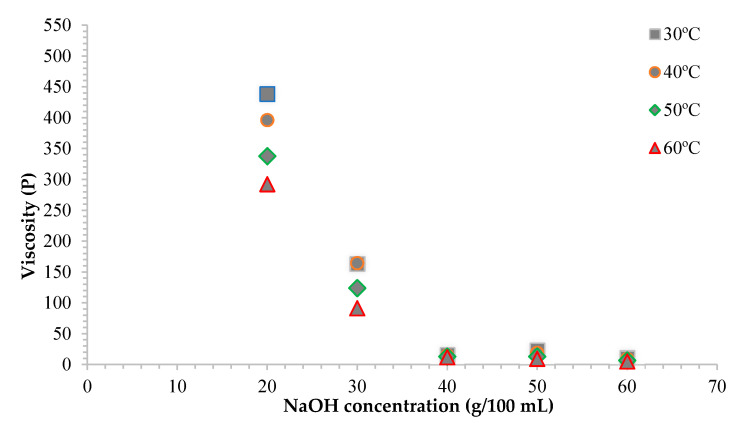
Effect of various NaOH concentrations on viscosity of CMC_as_.

**Figure 5 polymers-13-00081-f005:**
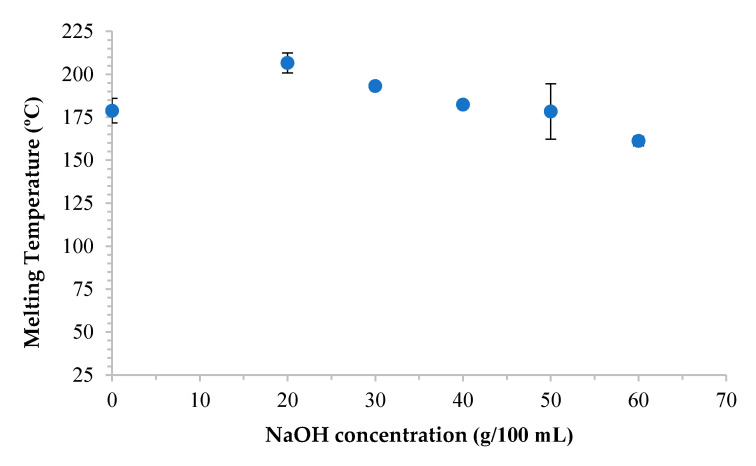
Thermal property of cellulose from asparagus stalk end and CMC_as_ synthesized with various NaOH concentrations.

**Figure 6 polymers-13-00081-f006:**
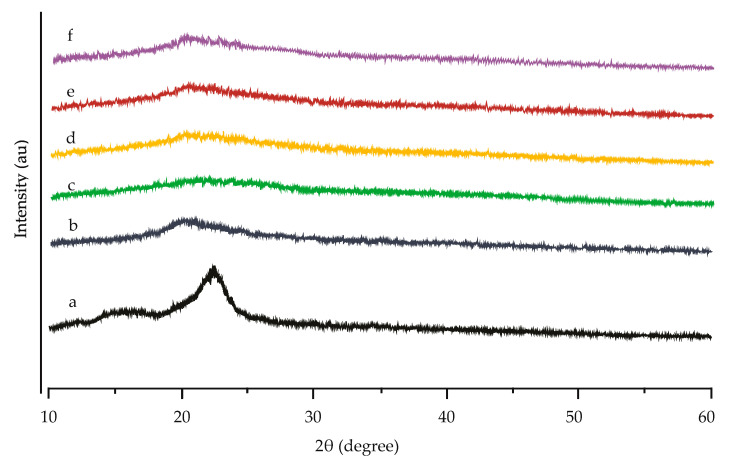
X-ray diffractograms of (a) cellulose from asparagus stalk end and CMC_as_ powder: with (b) 20% (*w*/*v*) NaOH, (c) 30% (*w*/*v*) NaOH, (d) 40% (*w*/*v*) NaOH, (e) 50% (*w*/*v*) NaOH and (f) 60% (*w*/*v*) NaOH.

**Figure 7 polymers-13-00081-f007:**
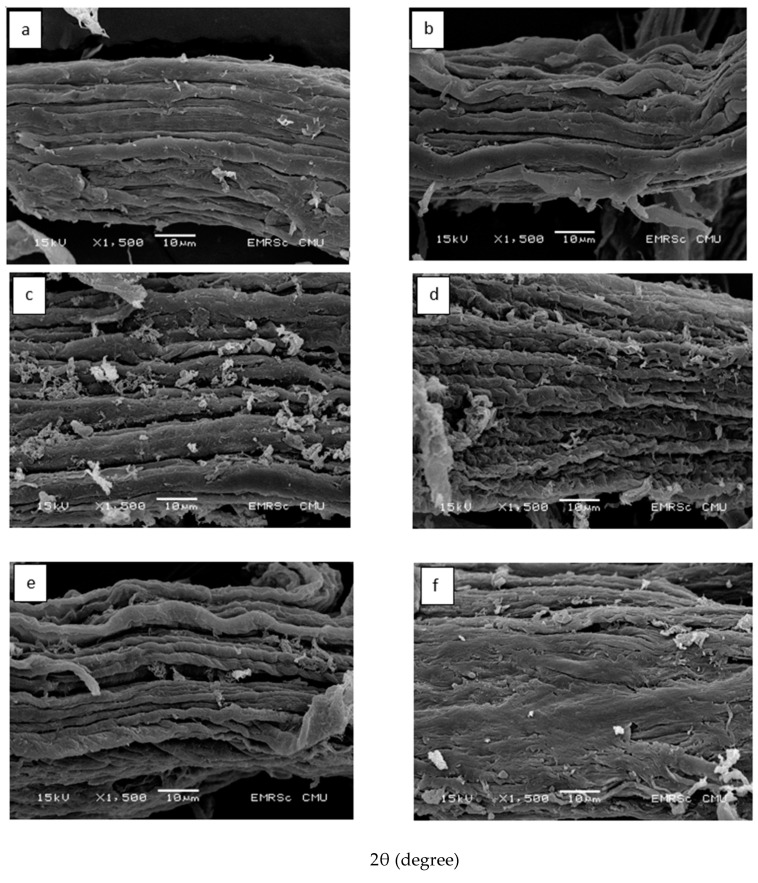
Scanning electron micrographs of (**a**) cellulose from asparagus stalk end and cmc_as_ powder: with (**b**) 20% (*w*/*v*) naoh, (**c**) 30% (*w*/*v*) naoh, (**d**) 40% (*w*/*v*) naoh, (**e**) 50% (*w*/*v*) naoh and (**f**) 60% (*w*/*v*) naoh. The acceleration voltage was 15 kv under low with 1500×.

**Table 1 polymers-13-00081-t001:** Color values of cellulose and CMC_as_ synthesized with various NaOH concentration.

Type of Sample	*L**	*a**	*b**	*WI*	*YI*	∆*E*	*H* _ab_
Cellulose	71.51 ± 0.37 ^b^	3.01 ± 0.05 ^c^	21.31 ± 0.26 ^b^	61.83 ± 0.33 ^b^	31.21 ± 1.89 ^c^	29.62 ± 0.24 ^ns^	5.87 ± 0.17 ^a^
20 g/100 mL NaOH-CMC_as_	73.88 ± 0.45 ^a^	2.69 ± 0.28 ^d^	20.77 ± 0.50 ^c^	64.54 ± 0.55 ^a^	45.02 ± 1.05 ^a^	37.46 ± 21.89	6.33 ± 0.90 ^a^
30 g/100 mL NaOH-CMC_as_	71.78 ±0.74 ^b^	2.87 ± 0.08 ^c,d^	21.70 ± 0.38 ^b^	61.99 ± 0.62 ^b^	45.02 ± 1.05 ^a^	29.67 ± 0.35	6.31 ± 0.10 ^a^
40 g/100 mL NaOH-CMC_as_	68.51 ± 0.44 ^c,d^	3.69 ± 0.25 ^b^	22.78 ± 0.17 ^a^	58.52 ± 0.42 ^d^	45.20 ± 1.01 ^a^	32.90 ± 0.32	4.94 ± 0.14 ^c,d^
50 g/100 mL NaOH-CMC_as_	69.30 ± 0.47 ^c^	3.54 ± 0.14 ^b^	21.56 ± 0.29 ^b^	59.79 ±0.35 ^c^	41.52 ± 1.66 ^b^	31.50 ± 0.44	5.06 ± 0.17 ^b^
60 g/100 mL NaOH-CMC_as_	67.97 ± 1.33 ^d^	4.00 ± 0.16 ^a^	21.28 ± 0.31 ^b^	58.34 ± 1.25 ^d^	40.17 ± 1.59 ^b^	32.43 ± 0.94	4.42 ± 0.22 ^d^

Mean ± Standard deviation values within a column in the same group followed by the different letters (a–d) are significantly different (*p* < 0.05). ns = not significant difference.

**Table 2 polymers-13-00081-t002:** Mechanical properties for CMCas films synthesized with various NaOH concentrations.

Type of Film	Tensile Strength (MPa)	Elongation at Break (%)
20 g/100 mL NaOH-CMC_as_	36.30 ± 1.32 ^c^	13.00 ± 7.22 ^c^
30 g/100 mL NaOH-CMC_as_	41.78 ± 3.28 ^b^	24.99 ± 3.79 ^a^
40 g/100 mL NaOH-CMC_as_	44.59 ± 1.73 ^a^	17.32 ± 4.21 ^b^
50 g/100 mL NaOH-CMC_as_	28.97 ± 2.06 ^d^	9.52 ± 2.57 ^c^
60 g/100 mL NaOH-CMC_as_	27.55 ± 1.72 ^d^	2.38 ± 0.88 ^d^

Mean ± Standard deviation values within a column in the same group, followed by the different letters (a–d) are significantly different (*p* < 0.05).

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
