# Peer review of "Synthesis, Characterization, and Application of Carboxymethyl Cellulose from Asparagus Stalk End"

_polymers, 2020, doi:10.3390/polym13010081_

Round 1
Reviewer 1 Report
In paper, synthesis and investigations on carboxymethyl cellulose have been presented. Topic of the research is interesting. The article is well-organized. The methodology of the research and the results of the experiments are well-described. The background of the study has been briefly presented in the Introduction of the article. In general, paper is worth considering for publication but some improvements are suggested. All of them are described more widely below:
- Abstract of the paper should be supplemented with few sentences concerning the significance of the research topic and its novelty.
- Sodium hydroxide is not adequate keyword, such as “carboxymethylation” or “cellulose extraction” is suggested.
- The process of the cellulose extraction from dried asparagus stalk end powder should be briefly described (Section 2.3.).
- Quality of Figure 3. should be significantly improved. Furthermore, it is suggested to present FT-IR spectra in different colors and describe as a, b, c etc. because they are poorly visible now. The same recommendations apply to Figure 6.
- Paper should be significantly re-checked grammatically and linguistically. Additionally, it contains some misspellings (e.g. “conetent” instead of “content” in equation 4.).
Reviewer 2 Report
Comments
The paper entitled “Synthesis, Characterization and Application of Carboxymethyl Cellulose from Asparagus officinalis Stalk End ” is an interesting article.
In this research article the authors proposed the extraction and synthesis of cellulose from Asparagus officinalis stalk end to carboxymethyl cellulose (CMCas) using monochloroacetic acid (MCA) via carboxymethylation reaction with various sodium hydroxide (NaOH) concentrations starting from 20% to 60 %.
The authors proposed several characterizations, the cellulose and CMC as were characterized by the physical, chemical, thermal and morphological properties. In addition, mechanical properties of CMC as films were also investigated. The authors studied that the optimum condition for producing CMCas was found to be used 30% of NaOH concentration for the carboxymethylation reaction, which provided the highest percent yield of CMC as at 44.04% with the highest degree of substitution (DS) at 0.98. The thermal analysis highlighted that the melting point of CMCas decreased with increasing NaOH concentrations. Crystalinity of CMC as was significantly modified after synthesis at a high concentration. The mechanical investigation permitted to determine that the highest tensile strength (44.59 MPa) was found in CMC as film synthesized with 40% of NaOH concentration and the highest percent elongation at break (24.99%) was obtained in CMC as film treated with 30% of NaOH concentration.
The paper is interesting, a lot of characterizations are presented in this work. I have no hesitation to suggest the publication of this manuscript after some revision. The manuscript will be published in Polymers Journal.
Specific comments
Introduction-paragraph 1: The authors are invite to stress better the novelty and the final application.
carboxymethyl cellulose (CMC) synthesized from asparagus stalk end-paragrapg 2.4: the authors are invited to use the capital letter for carboxymethyl.
Figure 2: The author are invited to eliminate the peak relative to CO2 present in all the spectra.
Effect of various NaOH concentrations on thermal properties of CMCas-paragraph 3.5: The authors are invited to insert the DSC thermograms.
Color of cellulose from asparagus stalk end and CMCas- paragraph 3.8: The authors are invited to improve the comment of this paragraph.
